# Antibacterial potential of *Luidia clathrata* (sea star) tissue extracts against selected pathogenic bacteria

**Kusum Parajuli, Nahian Fahim, Sinthia Mumu, Rebecca Palu, Ahmed Mustafa** * 

Department of Biological Sciences, Purdue University Fort Wayne, Fort Wayne, IN, United States of America

* mustafaa@pfw.edu

## Abstract

As resistance to traditional antibiotics has become a major issue, it is essential to explore natural sources for new antimicrobial agents. The marine environment offers a variety of natural bioactive compounds. In this study, we examined the antibacterial potential of *Luidia clathrata*, a tropical sea star species. The experiment was conducted against both gram-positive (*Bacillus subtilis*, *Enterococcus faecalis*, *Staphylococcus aureus*, *Bacillus cereus* and *Mycobacterium smegmatis*) and gram-negative (*Proteus mirabilis*, *Salmonella typhimurium*, *Escherichia coli*, *Pseudomonas aeruginosa* and *Klebsiella pneumoniae*) bacteria using disk diffusion method. Specifically, we extracted the body wall and gonad using methanol, ethyl acetate, and hexane. Our findings show that the body wall extract using ethyl acetate (1.78μg/ml) was particularly effective against all tested pathogens, while the gonad extract (0.107μg/ml) showed activity against six out of ten selected pathogens. This is a crucial and new discovery that suggests *L. clathrata* may be a useful source for discovering antibiotics and more research is required to pinpoint and comprehend the active ingredients.

**Data Availability Statement:** All the data are available within the paper and the Supporting Information file.

## 1. Introduction

Multi-drug resistance (MDR) pathogens that have arisen over the past decade are a considerable threat to patients' health [1,2]. These MDR microorganisms evolve through mutation and gene transfer in response to the prolonged use and misuse of certain drugs [1–3]. Numerous adverse side effects possessed by conventional antibiotics are another problem related to health [1,4,5]. It is crucial to develop a sustainable solution to mitigate the limitations of existing antibiotics. Exploration of nutraceuticals in natural sources could serve as an effective way to develop such a solution [3,6–8]. Marine environments are one potentially overlooked resource for nutraceuticals [9].

The oceans, which cover almost 70% of the earth's surface, offer a myriad of organisms rich in secondary metabolites that can be exploited for pharmaceutical purposes [9,10]. Secondary metabolites are organic compounds produced by plants and animals which are not essential for their survival and growth but are utilized in defense responses [11–13]. These compounds

**Funding:** The author(s) received no specific funding for this work.

**Competing interests:** The authors have declared that no competing interests exist.

include but are not limited to echinochrome A, complement-like protein, antimicrobial peptides (AMP), steroidal glycosides, asterosaponin, and sulfated steroidal compounds [14–19]. All have been previously isolated from the echinoderms and studied for their medicinal importance [14–19]. The results from those studies imply that such compounds have diverse medicinal properties including anti-microbial, anti-inflammatory, antioxidant, and anticancer effects [14–20].

The sea star is a keystone predator in marine ecosystems full of bioactivities and nutraceutical properties, but they have been poorly studied compared to other echinoderms such as sea cucumber, sea urchins, and brittle stars [21,22]. This benthic free-living creature is well documented for its distinctive defensive mechanism to mitigate the disadvantage and ecological cost associated with other commensal or parasitic surface associated organism. [17,23]. The surface microtopography of some tropical sea stars has demonstrated the presence of a unique cuticle overlying the epidermis. This cuticle is rich in highly extended glycocalyx and chondroitin sulfate proteoglycans, which are pericellular glycoproteins that cover the cell and act as a physical barrier [17]. These surface-associated bioactive compounds provide good protection from pathogens by modulating the adhesive properties of the surface [17,24]. Although existing research has well documented the bioactivity and pharmaceutical potential of various sea star species, *Luidia clathrata*, a tropical slender armed sea star, has been barely studied for its antibacterial potential.

In this experiment, we investigated the antimicrobial potential *of L. clathrata*. We analyzed the inhibitory properties of different body tissues (body wall and gonad) with respect to diverse pathogenic bacteria. We used three different solvents (methanol, ethyl acetate, and hexane) exhibiting different properties to extract different bioactive compounds from these tissues. We used the Kirby Bauer Disk Diffusion method to assess the inhibitory potential of extracted tissues [25]. The results showed that the body wall extracted with ethyl acetate possesses inhibitory properties across all tested pathogens, while gonad extract only inhibits the activity of a few pathogens. Methanol and hexane extracts did not produce any activity. Methanol extract of the body wall demonstrated hemolytic activity on red blood cells. This encouraging finding implies that the body wall and gonad of *L. clathrata* could serve as an important source of antibiotics for pathogenic bacteria.

## 2. Material and methods

### 2.1. Species acquisition and maintenance

24 Healthy sand sifting sea star adults (24.41±1.50gm) were procured from a certified animal vendor (Gulf Specimen Marine Lab, Panacea, Florida, USA). Upon arrival, the species were maintained in optimal water conditions (temperature: 68–70˚F, salinity: 28±1ppt, ammonia: 0–0.25mg/L, pH:7.8–8.0) in the invertebrate lab. The specimens were thoroughly cleaned with de-ionized water to remove any adherent sediments and contaminants before dissection. 24 sea stars were dissected to collect the body wall, and gonad. The different components were then pooled separately. Due to their fragility, the gonads were homogenized using a tissue homogenizer, while the body wall tissues were finely ground using a coffee grinder (Hamilton Beach® Fresh Grind™).

### 2.2. Preparation of extract

The extraction procedure was carried out by following the methods described by Shuchizadeh et al. with some modifications [26]. The gonad (5gm), and body wall (84.3gm) were submerged in reagent grade (99%) methanol, hexane, and ethyl acetate (PRA grade, ≤ 99.5%, Sigma-Aldrich) in 1:3 (w/v) ratio and constantly agitated on orbit shaker (Lab-line Orbit

Shaker, Model 3520) for 96 hours at room temperature. The flasks were covered with aluminium foil to avoid photolysis and thermal degradation of secondary metabolites prior to extraction. The extract was then decanted and filtered with Whatman® Grade 3 Filter Paper (diameter 12.5cm). The resulting filtrate was concentrated using a rotary evaporator (BU-R134 Rotary Vap System, Switzerland) at reduced pressure and temperature (40–45˚C). The concentrated crude residues were stored at 4˚C for the subsequent investigations.

## 2.3. Determination of crude extract concentration

The volume of concentrated crude extract was measured and transferred to the previously weighted empty dish. The total weight of the crude extract with the dish was taken. The concentration was calculated using the following formula [27]:

$$Concentration = \frac{(Weight_{extract+dish} - W_{empty\ dish})}{Volume\ of\ Crude\ extract\ in\ ml} x \frac{1000mg}{g}$$

## 2.4. Test microorganism and culture medium

Five gram-positive bacteria [*Bacillus subtilis*[X], *Enterococcus faecalis* (ATCC 25922), *Staphylococcus aureus* (ATCC 27659), *Bacillus cereus*[x] and *Mycobacterium smegmatis*[x]) and five gram-negative [(*Proteus mirabilis*[x], *Salmonella typhimurium* (ATCC 14028), *Escherichia coli* (ATCC 11229), *Pseudomonas aeruginosa* (ATCC 27853) and *Klebsiella pneumoniae* (ATCC 13883)] were examined in this experiment ([x] denotes that ATCC number is not available). All the bacteria, except *E. faecalis* were sub-cultured on Tryptic soy agar (TSA) media at 37˚C for 24hours. *E. faecalis* was grown on 5% sheep blood agar media. These subcultures were kept at 4˚C to guarantee bacterial viability and purity.

## 2.5. Antibacterial assay

Antibacterial activity was assessed by the disk diffusion method [25]. Petri plates (100mm and 150mm) were prepared by pouring 20ml and 60ml of Muller Hinton Agar (MHA) respectively. The plates were swabbed aseptically with fresh bacterial suspension prepared from the subculture maintained at 4˚C and standardized with 0.5 McFarland standard. A sterile filter paper disk (6mm) was impregnated with the extracted samples and placed on the agar surface along with positive and negative controls at an appropriate distance and incubated for 24hours at 37˚C. The extraction solvents were employed as negative controls, whereas antibiotics appropriate to the organism (gentamicin, vancomycin, penicillin, streptomycin, and SXT) were utilized as positive controls. The zone of inhibition was characterized by the formation of a clear zone around the disk. For the haemolytic activity, 5% sheep blood agar plate inoculated with *E. faecalis* was used. The zone of haemolysis was interpreted as a clear zone formed by destruction of red blood cells around the disk. The diameter of zone of inhibition and haemolysis were measured in millimetres.

## 2.6. Statistical analysis

The assays were maintained in triplicates and data obtained are presented as means± standard error of the mean (SEM). The assumption of normality was met. Comparison between negative control and sample extracted was performed by analysis of variance (ANOVA, $p < 0.05$) followed by Bonferroni correction.

**Table 1.** Antibacterial activity demonstrated by the ethyl acetate extracts of *Luidia clathrata* tissues (body wall and gonad) on selective pathogenic bacteria achieved by the disk diffusion method.

| Pathogens | Antibiotics (µg) | Zone of Inhibition (Diameter in mm) | | | |
|---|---|---|---|---|---|
| | | Positive control (Antibiotics) (Mean± SEM) | Negative Control (Ethyl acetate) (Mean± SEM) | Body wall (Mean± SEM) | Gonad (Mean± SEM) |
| **Gram Negative** | | | | | |
| *Proteus mirabilis* [x] | GM (10) | 27.33±1.33 | 0 | 34.00±0.88 | 12.00±0.66 |
| *Salmonella typhimurium* (ATCC 14028) | SXT (10) | 36.33±1.33 | 0 | 35.66±2.96 | 12.66±1.85 |
| *Escherichia coli* (ATCC 11229) | GM (10) | 30.00±0.55 | 0 | 34.66±1.45 | 12.33±1.33 |
| *Pseudomonas aeruginosa* (ATCC 27853) | ST (10) | 17.66±0.33 | 0 | 26.33±0.88 | - |
| *Klebsiella pneumoniae* (ATCC 13883) | GM (10) | 25.00±1.00 | 0 | 29.66±0.33 | 9.00±1.00 |
| **Gram Positive** | | | | | |
| *Bacillus subtilis* [x] | P (10) | 33.66±2.18 | 0 | 32.33±1.20 | 12±1.52 |
| *Enterococcus faecalis* (ATCC 25922) | GM (10) | 20.66±0.33 | 0 | 18.66±0.33 | - |
| *Staphylococcus aureus* (ATCC 27659) | P (10) | 38.33±0.33 | 0 | 37.66±2.33 | 11.33±0.33 |
| *Bacillus cereus* [x] | VA (30) | 19.00±0.00 | 0 | 20.66±0.33 | - |
| *Mycobacterium smegmatis* [x] | ST (10) | 28.00±1.72 | 0 | 44.66±2.90 | - |

[x] denotes that ATCC number is not available.

Values are presented as the mean diameter of inhibition zones (mm) ± standard error of the means (n = 3). GM (Gentamicin), SXT (Sulfamethoxazole-Trimethoprim), P (Penicillin), ST (Streptomycin) & VA (Vancomycin). '-' = no activity against the bacteria. All other interactions were significantly changed from the negative control ($p<0.05$).

# 3. Results

## 3.1. Ethyl acetate extracts exhibit broad-spectrum antibacterial activity

The antimicrobial activity of ethyl acetate extract of *L. clathrata* body wall (1.78µg/ml) and gonad tissues (0.107µg/ml) is summarised in Table 1. Ethyl acetate extract of the body wall exhibited significant antibacterial activity against all tested pathogens. Gonad extracted with ethyl acetate exhibited inhibitory activity against six out of the ten selected pathogens. Activity was not observed for the gonad extract against *Pseudomonas aeruginosa*, *Enterococcus faecalis*, *Bacillus cereus*, and *Mycobacterium smegmatis*. Overall antibacterial activity was also lower than that observed in the body wall extract. We did not observe the zone of haemolysis for any of the tissues extracted with ethyl acetate Table 2.

**Table 2.** Haemolysis activity of *L. clathrata* extract of body wall (1.78µg/ml) and gonad tissues (0.107µg/ml) extracted with different solvent by disk diffusion method.

| Solvent Used | Zone of Haemolysis (Diameter in mm) | | |
|---|---|---|---|
| | Control (solvents only) (Mean± SEM) | Body wall (Mean± SEM) | Gonad (Mean± SEM) |
| Methanol | 0 | 14±1.00 | - |
| Ethyl acetate | 0 | - | - |
| Hexane | 0 | - | - |

Results are illustrated as the mean diameter of haemolysis zones (mm)±standard error of the means (n = 3). '-' = no activity against the bacteria. All other interactions were significantly changed from the negative control ($p<0.05$).

## 3.2. Methanol and hexane extracts do not exhibit antibacterial activity

Methanol extract (1.78μg/ml) of any of the tissues exhibited no inhibitory activity against the selected pathogens. We did observe significant (p<0.05) beta-haemolysis, a complete destruction of red blood cells by the methanol extract of body wall Table 2. Because beta haemolysis was not observed in ethyl acetate extract, the responsible compound must be specifically soluble in methanol. Hexane extract of any of the tissues exhibited no inhibition against the selected pathogens. Because of the nonpolar nature of the hexane, any polar bioactive compounds would not be extracted [28]. Haemolytic activity was also not observed with the tissues extracted with hexane.

## 4. Discussion

The emergence of antibiotic resistant organisms has made treating the diseases they cause difficult [1]. Discovery of new therapeutic agents from natural sources could provide a potential solution. In this experiment, we aimed to determine the antibacterial activity of *L. clathrata* against selected pathogenic bacteria [2,3].

Existing literature has shown the wide range of bioactivity from a variety of marine invertebrates, but little information is available about the sea star antibacterial activity [11,14,23]. In our study, ethyl acetate extract of body wall showed a significant (p<0.05) zone of inhibition in all tested pathogens compared to the negative control. The zone of inhibition was highest against *M. smegmatis* (44.66±2.90mm) and smallest against *E. faecalis* (18.66±0.33mm). Our finding is supported by Bryan et al. [29]. They discovered body wall extract of *L. clathrata* that potentially inhibited the attachment of a marine bacteria *Luteo violaceato* from the wells of microtiter plates, indicating the defence mechanism of the body wall which could potentially be antibacterial in nature. However, they did not explain in detail the antimicrobial potential of the body wall [29]. Similarly, ethanolic extract of whole-body tissue from *L. maculatata* partially purified using liquid partition and column chromatography exhibited antimicrobial activity against five bacterial and five fungal pathogens [30]. Kanagaraj et al. studied the antibacterial activity of *Astropecten indicus* and found that crude methanol and ethyl acetate tissue extract exhibited high inhibitory activity against the tested pathogens including *P. aeruginosa*, *K. pneumoniae* and moderate activity against species like *Streptococcus* and *E. coli* [31]. In our case high activity was observed on all the tested pathogens. Previous research primarily focused on whole body tissues and the body wall [29,30]. In the present study, we have explored the antibacterial potential of the gonad as well for the first time along with the body wall. Gonad extracted in ethyl acetate was able to inhibit some of the tested pathogens. It is likely that the ethyl acetate extract of body wall was more effective than the gonad extract because of discrepancies in concentration. The concentration of body wall extract is about 16X higher than gonad extract. Another possibility could be due to the difference in the chemical nature of compounds present in two tissue type. This also explains the fact that gonad is likely more effective against the gram-negative pathogens compared to the gram-positive ones. Out of six pathogens being inhibited by gonad extract, four of them are gram-negative and two are gram-positive. The greater inhibitory activity against gram-negative pathogens could be because they have an extra lipopolysaccharides layer. Fatty compounds from gonads may dissolve the lipopolysaccharides and thus likely destroy gram-negative pathogens more readily than gram-positive [32].

The methanol extract of none of the tissues showed activity against tested pathogen. This is interesting because methanol is a widely used polar solvent due to its ability to extract a diverse range of compounds and proven to have good extraction yield [33,34]. However, we noticed beta-haemolytic activity of body wall extracted with methanol on 5% sheep blood agar. The

haemolytic activity by methanolic extract of body wall observed in the present experiment could be due to the presence of saponin in body wall [35,36].

Saponin, a polar secondary metabolite mostly found in plants and lower invertebrates is well characterized by its ability to breakdown red blood cells. This property is used as a screening test to determine whether saponin is present in natural substances [35,36].

In this experiment, the complete destruction of erythrocytes by the methanol extract of body wall suggests that body wall of *L. clathrata* is rich in saponin. The hexane extracts did not show any positive activity because hexane, as a non-polar solvent, is not able to extract the polar compounds present in the sample [37]. Since the ethyl acetate extract produced the majority of the positive results in the present studies, we anticipate that ethyl acetate is the proper solvent to extract the bioactive compounds with antibacterial nature from *L. clathrata*. Our results are in line with Darya et al., who reported the ethyl acetate extract of different body parts of *Holothuria leucospilota* and had more antibacterial activity than n-hexane, and methanol extract [38]. The present result of our study suggests that the antimicrobial compound(s) found in the body wall and gonad of *L. clathrata* is likely polar or partially polar.

## 5. Conclusion

In this research, we analyzed the antibacterial potential of *L. clathrata* tissues using diverse types of extracts of different polarities on selected pathogens. We found that ethyl acetate extracts of body wall and gonad tissues exhibit significant inhibitory activity. This indicates the studied species, *L. clathrata*, could be an excellent source for discovering antibiotics to treat various types of diseases. This work can be expanded through the isolation, characterization and purification of the specific compounds responsible for the antibacterial potential.

## Supporting information

**S1 File.**
(XLSX)

## Acknowledgments

We acknowledge the contribution of Arlis LaMaster, Laboratory Technician, Department of Biological Sciences, Purdue University Fort Wayne, Indiana, USA for her help during the experiment. Additionally, we appreciate Dr. Jaiyanth Daniel, Associate Professor, Department of Biological Sciences for his supervision during extraction.

## Author Contributions

**Conceptualization:** Kusum Parajuli, Nahian Fahim, Sinthia Mumu, Rebecca Palu, Ahmed Mustafa.

**Data curation:** Kusum Parajuli, Nahian Fahim.

**Formal analysis:** Kusum Parajuli.

**Investigation:** Kusum Parajuli, Nahian Fahim, Rebecca Palu, Ahmed Mustafa.

**Methodology:** Kusum Parajuli, Nahian Fahim, Sinthia Mumu, Rebecca Palu, Ahmed Mustafa.

**Project administration:** Ahmed Mustafa.

**Resources:** Ahmed Mustafa.

**Supervision:** Rebecca Palu, Ahmed Mustafa.

**Writing – original draft:** Kusum Parajuli.

**Writing – review & editing:** Kusum Parajuli, Nahian Fahim, Sinthia Mumu, Rebecca Palu, Ahmed Mustafa.

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
