## [Decision Letter · Decision Letter 0]

22 Dec 2022

PONE-D-22-30432Antibacterial Potential of Luidia clathrate (Sea Star) Tissue Extracts Against Selected Pathogenic BacteriaPLOS ONE

Dear Dr. Mustafa,

Thank you for submitting your manuscript to PLOS ONE. After careful consideration, we feel that it has merit but does not fully meet PLOS ONE’s publication criteria as it currently stands. Therefore, we invite you to submit a revised version of the manuscript that addresses the points raised during the review process.

We look forward to receiving your revised manuscript.

Kind regards,

Estibaliz Sansinenea

Academic Editor

PLOS ONE

Journal Requirements:

**Additional Editor Comments:**

The reviewers have serious concerns about the novelry and property of this work. The Ms should be revised carefully following the reviewers comments.

Reviewers' comments:

Reviewer's Responses to Questions

**Comments to the Author**

1. Is the manuscript technically sound, and do the data support the conclusions?

Reviewer #1: No

Reviewer #2: Partly

2. Has the statistical analysis been performed appropriately and rigorously? 

Reviewer #1: Yes

Reviewer #2: Yes

3. Have the authors made all data underlying the findings in their manuscript fully available?

Reviewer #1: Yes

Reviewer #2: No

4. Is the manuscript presented in an intelligible fashion and written in standard English?

Reviewer #1: No

Reviewer #2: Yes

5. Review Comments to the Author

Reviewer #1: * The manuscript is poorly written. It has plenty of grammatical errors as well as sentence-structure mistakes.

* The paper does not make a significant contribution to new knowledge in the discipline.

* The research idea lacks novelty.

Reviewer #2: Abstract: It is not clear which pathogens are tested in the abstract (line 35). It is worth mentioning the species names. In the methodological part of the abstract, the authors should specify which organs of sea star were used for the experiments. The abstract should clearly present the scope of the work and its effects.

Material and methods: How many organisms were purchased for research (line 99)? How long they were kept in the lab before the analysis (line 101-102)? Line 104-105: Which organs were taken for testing. It's not clear from these two lines: only body wall and gonad - or any other components too (gut)? It should be clear what type of materials were used for the investigation. Line 111: the purity and concentration of the extractants used and their manufacturer/supplier were not specified. Line 112: shaker – the mark and model should be added. Line 114: how long the decantation was carried out. Line 116: what was the pressure applied? Line 124-127: what was the origin of the bacteria used in the research, how they were propagated for the research? To determine the antibacterial activity of the extracts, the authors should perform chemical analyzes of these extracts. This data should be included in to research article. In the current state the results presented in the paper are too poor to be published.

Taking into account the above deficiencies in the description of the methodology, further review proceedings can be conducted after their completion.

6. PLOS authors have the option to publish the peer review history of their article (what does this mean?). If published, this will include your full peer review and any attached files.

Reviewer #1: No

Reviewer #2: No

---

## [Author Response · Author response to Decision Letter 0]

31 Jan 2023

PONE-D-22-30432

Antibacterial Potential of Luidia clathrata (Sea Star) Tissue Extracts Against Selected Pathogenic Bacteria

PLOS ONE

1. Response to Reviewer 1

We would like to thank the reviewer for the valuable comments/suggestions. We summarize our responses as follows.

Q1: Writing issues and grammatical errors.

Thank you for identifying these issues. We have fixed these issues (formatting, typos and grammatical errors) in the updated draft. 

Q2: Clarification on novelty of the research

Thank you for your comment. To the best of our knowledge, this is the first scientific work explaining the antimicrobial potential of different body tissues of Luidia clathrata against ten different pathogenic bacteria. Specifically, our effort to explore the therapeutic potential of gonads from the L. clathrata is unique. By sharing our results with the scientific community, we will be contributing to the advancement of this field and helping to spur future research efforts aimed at identifying the active compounds responsible for the observed antibacterial activity. We believe, our research is novel.

2. Response to Reviewer 2

We would like to thank the reviewer for the valuable comments/suggestions. We summarize our responses as follows. 

Q1: Abstract revision (Which pathogens were used in the experiment? Provide detailed names of the pathogens. Mention the tissues used in the experiment.)

We have done a major revision of the abstract in light of reviewer’s comments. Please refer to the highlighted lines throughout the abstract. 

Regarding the bacterial names, we have mentioned the bacterial species and strain in detail in the methodology section. However, as the reviewer wanted us to add the names in the abstract, we have now incorporated them. 

We also have specified tissues/organs used in the experiment- in the abstract now.

Q2: Clarification on Material and Methods

Thank you for your questions and pointing out deficiencies. We have incorporated all the issues, as suggested.

Briefly,

- We have added the sample size 

- We have mentioned the acclimation period 

- We have added the tissue/organs used in the experiment

- Regarding the purity and concentration of the extractants used, we have updated the information

- We have updated the detailed information about the instruments used.

In addition, we like to inform you that the decantation period was 5 minutes and the pressure for rotary evaporator was 713 mmHg at 40-45oC (Reference 1).

Q3: Clarification on the chemical analysis of the extracts used.

Thank you for your suggestion. This result is the first part of our experiment that we would like to share. We will follow up your suggestion on chemical analysis of the extracts used in our follow up research with standard methodology.

References:

1. Souza, C. R. F., Schiavetto, I. A., Thomazini, F. C., & Oliveira, W. P. D. (2008). Processing of Rosmarinus officinalis Linne extract on spray and spouted bed dryers. Brazilian Journal of Chemical Engineering, 25, 59-69. https://doi.org/10.1590/S0104-66322008000100008

---

## [Editor Report · Decision Letter 1]

2 Feb 2023

Antibacterial Potential of Luidia clathrata (Sea Star) Tissue Extracts Against Selected Pathogenic Bacteria

PONE-D-22-30432R1

Dear Dr. Mustafa,

We’re pleased to inform you that your manuscript has been judged scientifically suitable for publication and will be formally accepted for publication once it meets all outstanding technical requirements.

Kind regards,

Estibaliz Sansinenea

Academic Editor

PLOS ONE

Additional Editor Comments (optional):

The authors have followed all recommendations closely improving their MS, they have changed the abstract and made the pertinent clarifications, therefore the MS can be accepted in the current form.
---

## [Editor Report · Acceptance letter]

22 Feb 2023

PONE-D-22-30432R1 

Antibacterial Potential of *Luidia clathrata* (Sea Star) Tissue Extracts Against Selected Pathogenic Bacteria 

Dear Dr. Mustafa:

I'm pleased to inform you that your manuscript has been deemed suitable for publication in PLOS ONE. Congratulations! Your manuscript is now with our production department. 

Kind regards, 

on behalf of

Dr. Estibaliz Sansinenea 

Academic Editor

PLOS ONE